# Towards implementation of context-specific integrated district mental healthcare plans: A situation analysis of mental health services in five districts in Ghana

**Benedict Weobong**[1]*, **Kenneth Ayuurebobi Ae-Ngibise**[2,3], **Lionel Sakyi**[2], **Crick Lund**[4,5]

**1** Department of Social and Behavioural Sciences, College of Health Sciences, University of Ghana, Accra, Ghana, **2** Ghana Somubi Dwumadie (Ghana Participation Programme), East Legon, Accra, Ghana, **3** Kintampo Health Research Centre, Research and Development Division, Ghana Health Service, Kintampo North Municipality, Bono East Region, Kintampo, Ghana, **4** Centre for Global Mental Health, Health Service and Population Research Department, Institute of Psychiatry, Psychology and Neuroscience, King's Global Health Institute, King's College London, London, United Kingdom, **5** Alan J Flisher Centre for Public Mental Health, Department of Psychiatry and Mental Health, University of Cape Town, Cape Town, South Africa

\* bweobong@ug.edu.gh

**Data Availability Statement:** All relevant data are available on Figshare: https://figshare.com/s/dbe959556b722f737d93.

## Abstract

### Background

Access to quality mental health services in Ghana remains poor, yet little is known about the extent of the access gaps and provision of mental health services at the district level in Ghana. We aimed to conduct an analysis of mental health infrastructure and service provision in five districts in Ghana.

### Methods

A cross-sectional situation analysis was conducted using a standardised tool to collect secondary healthcare data, supplemented by interviews with key informants, across five purposively selected districts in Ghana. The Programme for Improving Mental Health Care (PRIME) situation analysis tool was adapted to the Ghanaian context and used for data collection.

### Results

The districts are predominantly rural (>60%). There were severe challenges with the provision of mental healthcare: there were no mental healthcare plans, supervision of the few mental health professionals was weak and unstructured, access to regular supplies of psychotropic medications was a major challenge, and psychological treatments were extremely limited given the lack of trained clinical psychologists. There were no available data on treatment coverage, but we estimate this to be <1% for depression, schizophrenia, and epilepsy across districts. Opportunities for mental health systems strengthening include: the commitment and willingness of leadership, the existence of the District Health Information

**Funding:** CL would like to acknowledge the funding that was provided to this project by the Foreign Commonwealth and Development Office (https://www.gov.uk/government/organisations/foreign-commonwealth-development-office) (award number: PO8604). The funders had no role in study design, data collection and analysis, decision to publish, or preparation of the manuscript.

**Competing interests:** The authors have declared that no competing interests exist.

Management System, a well-established network of community volunteers, and some collaboration with traditional and faith-based mental health service providers.

## Conclusion

There is poor mental health infrastructure across the five selected districts of Ghana. There are opportunities for strengthening mental health systems through interventions at the district healthcare organisation, health facility, and community levels. A standardised situation analysis tool is useful for informing district-level mental healthcare planning in low-resource settings in Ghana and potentially other sub-Saharan African countries.

## Introduction

The World Health Organization (WHO) recognises that people with mental health conditions and disabilities have the right to equitable access to high-quality treatment and support [1]. Mental health conditions are major contributors to the increasing burden of disease worldwide; they are the seventh leading cause of disability, contributing 4.9% of total disability-adjusted life years [2]. Three-quarters of this burden is spread among low- and middle-income countries (LMICs) [3]. In these settings, most people with mental health conditions go untreated. This is largely due to access challenges, which in Ghana have led to a treatment gap of 95%–98% [4].

Poor access to mental health services can be addressed. There is evidence of effective innovations to improve access through integrated care models, such as the WHO Mental Health Gap Action Programme (mhGAP) [5] and the Programme for Improving Mental Health Care (PRIME) [6]. These innovations have been largely successful, but there is still a need to understand how to scale up mental health services effectively, through integrating mental health into primary healthcare (PHC) [7].

Cohen's work highlights how previous attempts to integrate mental healthcare into PHC in LMICs have been fraught with challenges, particularly with respect to implementation and sustainability [8]. Only about a third of the countries included in the 2020 WHO *Mental Health Atlas* have been able to integrate mental health into PHC [9]. PRIME has demonstrated ways to tackle these challenges and the enormous impact such interventions have on patient outcomes [10–13]. For example, the PRIME studies showed very clearly the importance of research to enhancing our understanding of the PHC system and opportunities for integrating mental health services before implementation and scale-up of services.

Ghana Somubi Dwumadie (Ghana Participation Programme) is a UK Aid-funded four-year programme (2020–2024) designed to help contribute the needed research evidence for the scale-up of high-quality and accessible mental health services in Ghana [14]. The Programme adopts a systematic approach that includes the conduct of baseline situation analysis. A similar situation analysis was conducted previously in Ghana but it did not include the broader community and health service context [15].

Situation analysis is a useful approach to documenting a clear, detailed, and realistic picture of the opportunities, resources, challenges, and barriers, particularly when planning the integration of mental health into primary care. A situation analysis method and related instrument were developed as part of PRIME for five LMICs (Ethiopia, India, Nepal, South Africa, and Uganda) [16]. However, little is known about how a standardised situation analysis can be conducted across diverse districts in Ghana, and how this might inform the scale-up of mental

health services in Ghana. The lessons from this process are transferable and can inform other African countries that may wish to conduct situation analyses to scale-up mental health services.

The primary objective of this study was to conduct a situation analysis of mental health infrastructure and service provision in five districts in Ghana to inform the development and implementation of district mental healthcare plans, as part of the Ghana Somubi Dwumadie programme.

## Materials and methods

### Design

The study was a cross-sectional situation analysis that drew on secondary routine healthcare data supplemented by interviews with key informants. The study was conducted between November 2020 and March 2021.

### Setting

The study was conducted in five districts. The districts were selected by key mental health stakeholders in Ghana as potential demonstration sites for the implementation of mental healthcare plans as part of the Ghana Somubi Dwumadie. The districts are Anloga in the Volta Region, Ahanta West in the Western Region, Asunafo North in the Ahafo Region, Tolon in Northern Region, and Bongo in the Upper East Region (Fig 1). There are 16 administrative regions in Ghana, which are subdivided into 216 districts to ensure equitable resource allocation and efficient, effective administration at the local level.

The selection of the demonstration districts was guided by a set of criteria outlined in Ghana Somubi Dwumadie's framework for implementing district mental healthcare plans, ratified in 2020 in a high-level stakeholder engagement meeting in Accra, Ghana. Thirty-three participants attended the meeting, made up of regional health directors and leadership of Ghana Health Service (GHS) mental health unit, Department of Social Welfare, Mental Health Authority (MHA), Christian Health Association of Ghana, Ministry of Health, Ghana Education Service School Health Programme, and Ministry of Gender, Children and Social Protection. The criteria for district selection were finalised and approved at a meeting with key stakeholders from the MHA and GHS and were drawn from lessons from PRIME. The criteria included: geo-political equity (i.e. the five districts should be spread across the three zones of Ghana: southern, middle belt, and savannah); representativeness (i.e. the districts should be neither over-resourced nor under-resourced, particularly in relation to human resources, so that lessons could be generalised to inform scale-up in other districts); an appropriate level of prevailing mental health activity (i.e. there should be no existing sites for mental health research and no previous or on-going district mental health plans or national academic centres for mental health, as these would create an unrealistic environment that is not representative of other districts); and willingness of district leadership to engage.

### Study participants and secondary data sources

Target respondents across the districts included: district health directors, health administrators, health information officers, public health nurses and district mental health focal persons. A variety of sources contributed data for the situation analysis and the compilation of this report. These included: annual health reports for each district, covering 2019 and 2020; mental health service delivery statistics reported in the District Health Information Management System (DHIMS) for 2020; medium-term development plan for each district; the 2010 Population

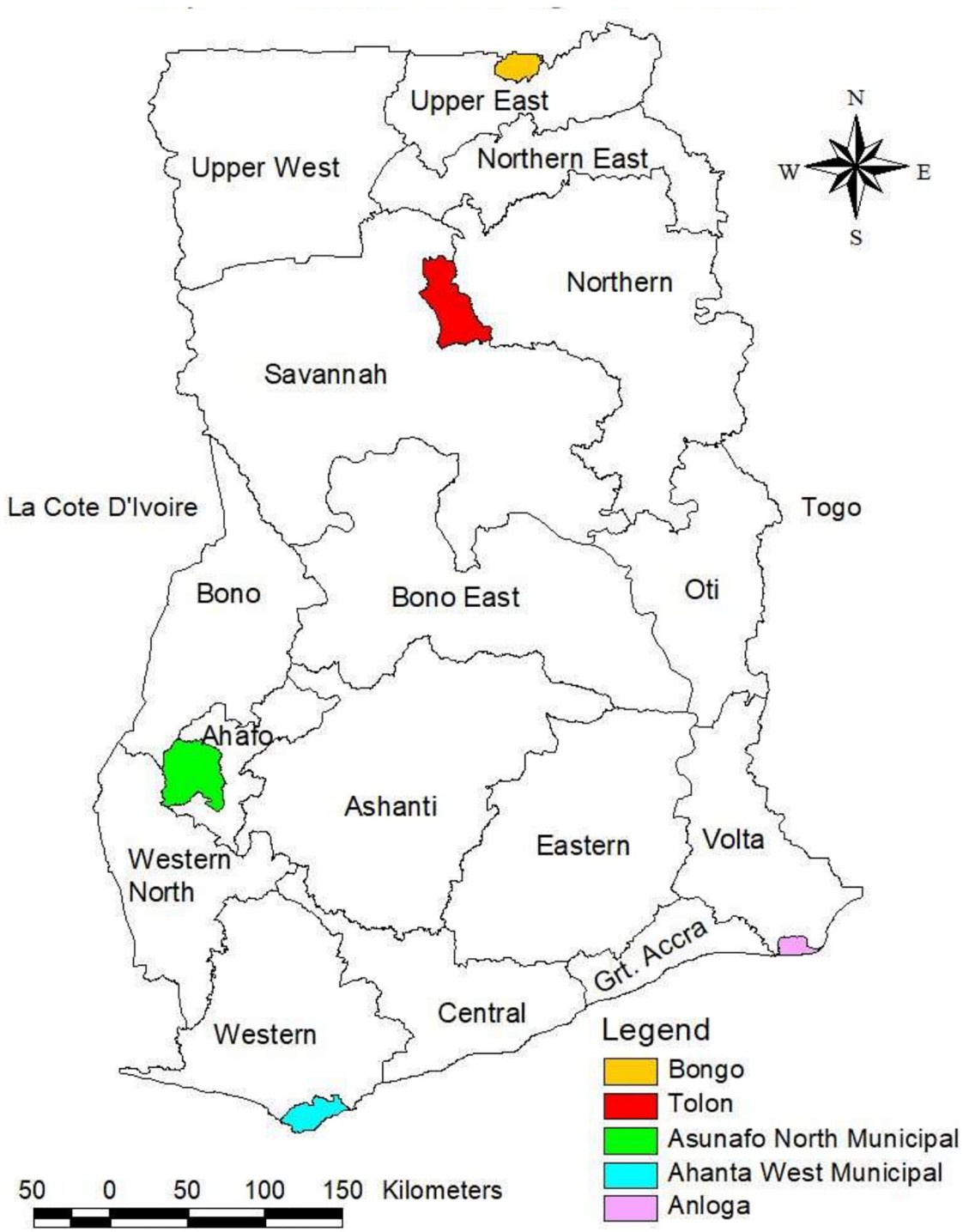

**Fig 1. Map of Ghana showing the five districts that participated in the study.**

and Housing Census (Ghana has a more recent 2020 census but this was not available at the time of data collection for this study); and district analytical reports.

## Ethical considerations

Ethical approval for the study was obtained from the Ghana Health Service Ethics Review Committee (GHS-ERC025/08/20) and the King's College London Research Ethics Committee (LRS-20/21-20866). Only participants who provided informed consent were included. Participants were assured of their rights to withdraw and confidentiality.

## Data collection tools and procedures

The PRIME situation analysis tool [16] was adapted to the Ghanaian context and used for data collection. This is a semi-structured tool used to collect key data on the context, challenges, and opportunities in providing mental health services. The adaptations were minimal and involved adapting key variables to the Ghanaian service delivery context, including the Community Health Planning and Services (CHPS) compounds as a type of health facility in Ghana. Details of the tool are provided elsewhere in the main report by Hanlon and colleagues [16], but in summary the tool comprises six sections with 156 items. The sections are: Relevant context; Mental health politics, policies and plans; Mental health treatment coverage; District level health services; Community; and Monitoring and evaluation. These sections are completed using primary data from respondent interviews or secondary data from government reports or research publications.

The challenges posed by Covid-19 were mitigated by ensuring strict observance of Covid-19 containment protocols (handwashing, use of face masks, maintaining a safe physical distance) for all in-person interviews.

## Data quality management

The adapted situation analysis tool was completed by an experienced local researcher (LR) familiar with the local context. Training on the tool and its administration was conducted by an author of the PRIME situation analysis tool, who is also an investigator in this study (CL). Data were recorded directly in the relevant sections of the tool, checked for accuracy, and thereafter entered on an Excel sheet for data management and analysis. Challenges with the accuracy and completeness of routine data were tackled through a report validation workshop, where participants provided critical clarification on the veracity of the data and sources.

## Data processing and analyses

Narrative data from reports and personal communications were analysed thematically, and quantitative data were reported as numbers, frequencies, or proportions and were disaggregated where such data were available. Data on treatment coverage (proportion of people with mental, neurological, and substance use (MNS) disorders in the districts accessing care) was challenging to assess because the parameters are not routinely assessed and reported. Given the importance of the treatment coverage indicator, we were only able to extrapolate using prevalence estimates for Ghana from the 2019 Global Burden of Disease study [17], the population of the district, and the total number of registered patients with mental health conditions for the top-three mental health conditions. The analysis also included comparison of the findings between districts. The comparison was useful in identifying opportunities and constraints in integrating mental health services into PHC in the districts. Following completion of analysis, a one-day online report validation workshop was convened with representatives from all

**Table 1. Selected socio-demographic characteristics by situation analysis district.**

| Indicator | Tolon | Bongo | Ahanta West | Asunafo North | Anloga* |
|---|---|---|---|---|---|
| District population (2020 projection) | 92,072 | 105,206 | 141,344 | 157,732 | 99,996 |
| % rural | 88.4% | 99.9% | 70.5% | 60.0% | 46.7% |
| Women in fertility age | 21,645 (46.5%) | 25,249 (24.0%) | 35,302 (25.0%) | 38,980 (24.7%) | 24,004 (23.0%) |
| Fertility rate | 3.2 | 4.0 | 3.9 | 3.9 | 3.8 |
| Sex ratio (female:male) | 100:99 | 100:90 | 100:92 | 100:102 | 100:87 |
| Average household size | 9 | 5.5 | 4 | 4.6 | 3.8 |
| Ethnic diversity | 1 major (98.2%) | 2 major (94.3%) | 1 major | 1 major (79.0%) | 1 major (98.7%) |
| Religious diversity | | | | | |
| Christianity | 3.7% | 28.3% | 78.6% | 77.8% | 59.9% |
| Islam | 94.1% | 6.4% | 3.2% | 14.9% | 1% |
| Traditional | 1.5% | 53.6% | 1.6% | 0.7% | 25.4% |
| No religion | 0.5% | 11.7% | 15.9% | 5.9% | 12.9% |
| Literacy (% who can read and write English and Ghanaian language only) | 73.8% | 33.2% | 50.6% | 70.8% | 75.1% |
| % of households with electricity | 32.9% | 11.5% | 70.9% | 93.1% | 41.8% |
| % of households with a functioning latrine | 10.5% | 20% | 20.3% | 38.6% | 57.0% |
| % of households with clean water supply | 49.0% | 70% | 60.1% | 72.7% | 79.6% |

* Except district population and women in fertility age, indicators for Anloga are based on information for Keta Municipal District. There is limited information on Anloga because the district was created in 2019 by the division of Keta District.

five districts, including district health directors, health administrators, health information officers, public health nurses, and district mental health focal persons, to review and ascertain the accuracy of the findings in the report.

## Results

### District contexts

**Socio-demographic and socioeconomic characteristics.** Table 1 presents information on residence, literacy, fertility, household size, religion, and utilities in the five districts. Overall, four out of the five districts are predominantly rural, with more than 60% of people living in rural areas. Literacy levels varied across the districts; relatively high (more than two-thirds of the population) in three districts (Tolon, Asunafo North, and Anloga), and low in the other two districts (range: 32.2%–50.6%). Fertility rate is similar across the districts (average: 3.7; range: 3.2–4.0), although the district with the lowest fertility rate (Tolon) also has the highest average household size (9 persons). Christianity is the dominant religion across the districts, except in Tolon and Bongo, which have predominantly Muslim and traditional religious faiths, respectively. We observe a diverse range of infrastructure across districts, in terms of access to electricity (range of access: 11.5%–93.1%) and sanitation (latrines and clean water; range of access: 10.5%–57%), although the two districts in the middle belt (Asunafo) and southern (Ahanta West) zones of Ghana have relatively better indicators, with more than two-thirds of households connected to the national electricity grid.

**General and PHC context for integrating mental healthcare.** In each district, the provision of general PHC services is overseen by the District Health Management Team, headed by a health director. PHC services are delivered at the district level through district hospitals, at sub-district level through health centres, and at community level through CHPS compounds. In four of the districts, all sub-districts are served by at least one health centre; the exception is

**Table 2. Distribution of health facilities, human resources, and common diseases by district.**

| Indicator | Tolon | Bongo | Ahanta West | Asunafo North | Anloga |
|---|---|---|---|---|---|
| **Number and type of health facilities** | | | | | |
| Hospital | 1 (just commissioned not functional) | 1 | 1 | 4 (1 govt, 1 CHAG, 2 private) | 0 |
| Clinic/community clinic with a medical officer | 2 (1 private, 1 UDS) | 0 | 2 | 0 | 0 |
| Health centres | 4 govt | 7 (6 govt, 1 CHAG) | 5 govt | 11 (1 CHAG, 7 govt, 3 private) | 6 govt |
| CHPS compounds | 14 | 22 | 36 | 11 | 4 |
| Outreach points | 120 | 69 | 110 | 78 | Not provided |
| Other health facilities | 0 | 62 | 0 | 0 | 0 |
| Hard to reach areas for health service delivery | Yes (36 communities) | Yes (3 communities) | None | Yes (13 communities) | None |
| **Human resources for health (general health workers)** | | | | | |
| Medical officers | 0 | 2 | 5 | 8 | 0 |
| Public health nurses | 1 | 1 | 1 | 2 | 0 |
| Midwives | 20 | 46 | 50 | 91 | 13 |
| Community health nurses | 40 | 111 | 98 | 67 | 27 |
| Staff/enrolled nurses (degree, diploma, certificate) | 135 | 253 | 128 | 134 | 54 |
| Physician assistants (medical) | 2 | 11 | 6 | 8 | 3 |
| Physician assistants (anaesthetics) | 0 | 3 | 0 | 0 | 0 |
| Health information officers | 2 | 4 | 3 | 6 | 1 |
| Disease control officers (TO/FT) | 5 | 8 | 3 | 10 | 3 |
| Nutrition officers | 5 | 5 | 1 | | 1 |
| Health promotion officers | 2 | 2 | 2 | 0 | 1 |
| Accountant | 2 | 2 | 2 | 2 | 1 |
| Ophthalmic nurse | 1 | 2 | 0 | 0 | 1 |
| Staff total | 215 | 450 | 299 | 328 | 105 |
| **Top-five reasons for outpatient visits** | Malaria (all categories) | Malaria (all categories) | Malaria (all categories) | Malaria (all categories) | Malaria (all categories) |
| | URTI | URTI | URTI | Diarrhoea diseases | Rheumatism / other joint pains / arthritis |
| | Diarrhoea diseases | Diarrhoea diseases | Anaemia | URTI | Anaemia |
| | Rheumatism / other joint pains / arthritis | Typhoid fever | Diarrhoea diseases | Anaemia | URTI |
| | Acute urinary tract infection | Septicaemia | Intestinal worms | Intestinal worms | Intestinal worms |
| MNS disorders in top-ten reasons for outpatient visits | No | No | No | No | No |

CHAG = Christian Health Association of Ghana; CHPS = Community Health Planning and Services; govt = government; MNS = mental, neurological and substance use disorders; TO/FT = technical officer/field technician; UDS = University for Development Studies; URTI, urinary and reproductive tract infections.

Tolon, where only four out of the six sub-districts have a health centre. All districts have CHPS compounds. Four districts have functional district hospitals (the exception being Anloga), and of these only Bongo and Ahanta West have medical officers in post in the hospitals. As shown in Table 2, the districts have a wide range of health professionals, including health promotion officers, nurses, midwives, physician assistants, and medical officers. The numbers are below the recommended Sustainable Development Goals index value of 4.45 skilled health workers

per 1000 population, or the global average of approximately 6.2 health workers per 1000 population [18].

An examination of the top-five reasons for outpatient department (OPD) attendance shows that the top-two reasons in Tolon, Bongo, and Ahanta West were similar. Malaria was the most common reason in all districts (range across districts: 32%–56%). MNS conditions did not feature in the top-ten reasons for OPD attendance in any of the districts.

## Mental health service provision

**Mental health services organisation.**   Ghana's Mental Health Act, 2012 (Act 846) [19] clearly outlines the establishment of the Mental Health Authority (MHA), the core mandate of which is regulatory and policy formulation. Mental health services are thus, according to policy and legislation, provided at all levels of the healthcare delivery system in Ghana. The services are jointly provided by the MHA and the Ghana Health Service (GHS). In addition to its regulatory role, the MHA is also directly responsible for all tertiary mental health services in the three psychiatric hospitals in Ghana. These psychiatric hospitals are in the southern zone of Ghana. The GHS on the other hand is mainly responsible for mental health service delivery at the secondary and primary care tiers, coordinated by a directorate within the mental health unit of the GHS.

None of the five districts in this study were covered by a district mental healthcare plan. This notwithstanding, all the districts do have mental health services. The services in each district are coordinated by the District Mental Health Focal Person (one per district, usually a registered mental health nurse rank) with the assistance of sub-district coordinators. The focal person role is a temporary arrangement (pending the official appointment of District Mental Health Co-ordinators as stipulated in the Mental Health Act, 2012 (Act 846)) put in place by the MHA through its Regional Mental Health Co-ordinators to support the organisation of mental health activities at the district level. Across the five districts, the focal person role is quite diverse—sometimes a management/coordinating role, sometimes mainly clinical— which has implications for effective coordination of mental health services.

**Mental health professionals and mental health services at health facilities.**   Table 3 shows the existing human resources for mental health services provision in the five districts. Community mental healthcare is largely provided by community mental health workers. There are four main groups within this cadre of mental health professionals across all districts in Ghana [20]: community mental health officers (12 months' full-time training); community psychiatric nurses (36 months' full-time training; community based); registered mental health nurses (36 months' full-time training); and clinical psychiatric officers (24 months' full-time training plus 1–3 months' 'internship' in a specialist hospital). Across all five districts, there is evidence of limited availability of these mental health professionals, including that there are no clinical psychologists or occupational therapists. There is wide variation in human resource availability for mental health services provision across the districts. For example, in Tolon, one mental health professional serves a population of 18,414, while the ratio in Bongo is one to 892, and one to 14,895 in Anloga.

Mental health services are provided at the district hospitals (except in Anloga), health centres, and CHPS compounds (Table 3). Services are predominantly on an outpatient basis, and these are offered at the health centre level and in the district hospitals (only in Bongo, Ahanta West and Asunafo North). The outpatient service provision at the district hospital and health centre levels is usually led by mental health professionals. While it is a generally accepted practice for primary care physicians and other prescribers to diagnose mental health conditions

**Table 3. Organisation of mental health services, mental health professionals and type of mental health services by district.**

| | Tolon | Bongo | Ahanta West | Asunafo North | Anloga |
|---|---|---|---|---|---|
| **Mental health service organisation** | | | | | |
| District mental health plan or implementation of national mental health plan | No district mental health plan | | | | |
| | Implementation based on routine service and activities determined and funded nationally | | | | |
| | Some mental health activities added to annual plans of the district health directorate | | | | |
| Budget for mental health (% of district health budget) | No specific district budget allocated to mental health | | | | |
| | Activity budgets from regional through national and MHA | | | | |
| Mental health coordinator | Yes, but known as district mental health focal persons based at district and sub-district levels | | | | |
| | The district mental health focal person is usually part of the District Health Management Team | | | | |
| Mental health is part of PHC basic packages | Yes | Yes | Yes | Yes | Yes |
| | Implementation not comprehensive | Good progress with implementation, e.g. some staff trained in mhGAP | Limited to facilities that have MHPs | Limited to the facilities that have MHPs | Limited to facilities that have MHPs |
| Screening tools for MNS disorders | Available | Available | Not available | Not available | Available, but limited, not enough |
| | Midwives and some CHNs have tool for depression screening | Depression screening tool | MHP usually go to the internet to search | Use what was learnt in school | |
| **Personnel type** | | | | | |
| Neurologists | 0 | 0 | 0 | 0 | 0 |
| Psychiatrists | 0 | 0 | 0 | 0 | 0 |
| Clinical psychiatric officers | 1 | 1 | 0 | 0 | 0 |
| Psychiatric nurses | 4 | 0 | 10 | 9 | 3 |
| Community mental health officers | 2 (on study leave) | 5 | 0 | 1 | 1 |
| Clinical psychologists | 0 | 0 | 0 | 0 | 0 |
| Occupational therapists | 0 | 0 | 0 | 0 | 0 |
| Community psychiatric nurses | 0 | 0 | 0 | 0 | 3 |
| Counsellors | 0 | 0 | 0 | 0 | 0 |
| Registered mental health nurses | 0 | 7 | 0 | 8 | 0 |
| Others | 0 | 0 | 0 | 0 | 0 |
| In-service training on mental health | 24 CHNs and 3 midwives trained in depression management | 7 staff trained (cannot remember content) | None in the last 2 years | 5 days mhGAP training (7 staff) | No training in the last 2 years |
| | 12 staff trained in mental health reporting | | | | |
| **Availability/type of mental health services** | | | | | |
| Outpatient mental health facilities | All health centres provide basic diagnosis and treatment of minor cases | Six health centres and district hospital provide diagnosis and treatment of minor cases | Basic diagnosis and treatment provided in district hospital and all sub-districts | District hospital and six out of 11 health centres with MHPs provide basic diagnosis and treatment of cases | Only two health centres provide mental health services |
| | Three health centres with MHPs provide more detailed services | | | | |
| Psychosocial interventions | Across districts, psychosocial interventions offered include: supportive counselling; behaviour activation, cognitive behaviour, and motivation enhancing therapies | | | | |
| Alcohol detoxification | Not offered | Yes | Not offered | Yes (able to hospitalise for 2 to 3 days if no improvement then refer) | Not offered |

*(Continued)*

**Table 3.** (Continued)

| | Tolon | Bongo | Ahanta West | Asunafo North | Anloga |
|---|---|---|---|---|---|
| Mental health rehabilitation | Not offered | Not offered | Not offered | Not offered | Not offered |
| Inpatient services in the district | No inpatient services | Yes (no dedicated beds) | Yes (no dedicated beds) | Yes (no dedicated beds) | No inpatient services |
| Nearest mental health inpatient facility | Tamale Teaching Hospital (22km away) | Regional hospital (15km); private facility in the regional capital (15km); Tamale Teaching Hospital (175km away); | Ankaful Psychiatric Hospital (about 160km away) | Sunyani Regional Hospital (84km away) | Accra Psychiatric Hospital(165km) or Pantang Psychiatric Hospital (172km away) |
| Specialist inpatient for alcohol detoxification | Accra or Pantang Psychiatric Hospitals | | Ankaful Psychiatric Hospital | | Accra or Pantang Psychiatric Hospitals |
| Outreach services for patients with severe mental illness (SMI) | SMI treated at home with the support of families | SMI treated at home with the support of families | SMI treated at home and on OPD basis | Patients stabilised at hospital and treatment continued at home | SMI treated at home |
| | Families educated to take care of patients so that they do not leave home | Families educated to take care of patients so that they do not get lost | Families educated to keep patients at home | Families counselled on importance of family support | Patients may come in for OPD services once in a while |
| | Families supported by community volunteers trained under the WHO 'Fight against epilepsy' initiative | | Many patients lost to follow-up because of few follow-up visits | Outreach services are generally limited because of lack of logistics and funding for mental health activities | Families counselled to help keep patients at home |
| | | | MHPs use their own money for transport when going on visits | The few outreach services are provided by MHPs at the sub-district level | |
| | | | A lot of SMI relapse when medication is used up and patients are unable to afford more | | |

CHN = community health nurse; MHA = mental health authority; mhGAP = Mental Health Gap Action Programme (WHO); MHP = mental health professional; MNS = mental, neurological, and substance use; OPD = outpatient department; PHC = primary healthcare; SMI = severe mental illness.

and prescribe psychotropic medication, the evidence from this study suggests that this is not being done and suspected cases are asked to consult the mental health professionals directly.

At the health centre and sub-district levels, suspected cases are usually referred first to the mental health nurses and subsequently to the district mental health focal person. There is some evidence of integrated mental health care. For example, in Tolon and Bongo, midwives provide depression (both antenatal and postnatal) screening and services and community health nurses support with health education. Furthermore, in all districts, mental health service users go through the general OPD in the same way as everybody else, starting with a routine consultation with a general physician (including examination for physical health conditions) before referral for specialist mental health services. Routine recorded data on psychosocial services are quite patchy, but what are available suggest the availability of services varies between districts. Mental health nurses are able to provide supportive counselling, including the use of other psychological therapies, such as cognitive behavioural therapy and motivation enhancing therapy. Approximate figures for the most recent year (2020) are that 40 patients in Bongo had an average of two sessions per year, 50 patients in Tolon had eight sessions, 56 patients in Asunafo North had three sessions, 17 patients in Anloga had two sessions, and 50 patients in Ahanta West had two sessions. There were, however, competency challenges because of lack of both supervision and opportunities for continuous training. There were also no mental health rehabilitation services in the districts. In terms of other support systems for mental healthcare,

**Table 4. Actual and population standardised newly registered patients in the most recent year (2020) by district.**

| MNS disorder | Tolon | | Bongo | | Ahanta West | | Asunafo North | | Anloga | |
|---|---|---|---|---|---|---|---|---|---|---|
| | New cases | Per 100,000 population | New cases | Per 100,000 population | New cases | Per 100,000 population | New cases | Per 100,000 population | New cases | Per 100,000 population |
| Schizophrenia | 07 | <0.1 | 79 | 0.25 | 19 | <0.1 | 10 | <0.1 | 69 | 0.23 |
| Depression | 02 | <0.1 | 36 | 0.11 | 12 | <0.1 | 08 | <0.1 | 03 | <0.1 |
| Epilepsy | 43 | 0.15 | 80 | 0.25 | 29 | <0.1 | 28 | <0.1 | 50 | 0.16 |
| Bipolar | 0 | <0.1 | 02 | <0.1 | 0 | <0.1 | 08 | <0.1 | 03 | <0.1 |
| Mental disorders due to alcohol use | 0 | <0.1 | 20 | <0.1 | 16 | <0.1 | 08 | <0.1 | 07 | <0.1 |
| Mental disorders due to substance use | 0 | <0.1 | 16 | <0.1 | 06 | <0.1 | 04 | <0.1 | 11 | <0.1 |
| Delirium | 0 | <0.1 | 04 | <0.1 | 0 | <0.1 | 36 | <0.1 | 01 | <0.1 |
| Dementia | 0 | <0.1 | 01 | <0.1 | 05 | <0.1 | 09 | <0.1 | 03 | <0.1 |
| GAD | 0 | <0.1 | 09 | <0.1 | 05 | <0.1 | 04 | <0.1 | 03 | <0.1 |
| Intellectual impairment | 0 | <0.1 | 05 | <0.1 | 01 | <0.1 | 10 | <0.1 | 01 | <0.1 |
| Other mental disorders | 0 | <0.1 | 06 | <0.1 | 26 | <0.1 | 67 | 0.14 | 06 | <0.1 |

GAD = generalised anxiety disorder.

there were no mental health detection screening tools or approved training manuals on mental health conditions.

**Mental health services uptake and coverage.** Assessment of data on treatment coverage (proportion of people with MNS disorders in the districts accessing care) was challenging. As shown in Table 4, the actual and population standardised numbers of newly registered patients with mental health conditions presenting at health facilities in each of the five districts are available for the most recent year (2020), but the numbers are quite unreliable. Overall, the most commonly diagnosed mental health conditions at OPD are epilepsy, schizophrenia, and depression. In Asunafo North, however, patients were most frequently recorded as presenting with non-specified mental health conditions. Mental disorders due to alcohol use were recorded in all districts except Tolon; the high Muslim population in Tolon could have contributed to this as Islam prohibits the consumption of alcohol.

Table 5 shows an estimate of treatment coverage and the population prevalence of the top-three mental health conditions in the most recent year. Given the importance of the treatment coverage indicator, we were only able to extrapolate using prevalence estimates for Ghana from the 2019 Global Burden of Disease study (schizophrenia: 0.2%; depression: 3.71%; epilepsy: 0.43%) [17)], the population of the district, and the total number of registered patients with mental health conditions in each district. Based on this, at <1% the overall crude treatment coverage for depression, schizophrenia, and epilepsy across districts is quite low (Anloga: 0.15%; Bongo: 0.55%; Asunafo North: 0.09; Ahanta West: 0.21; Tolon: 0.58).

**Availability and sources of psychotropic medication.** Table 6 shows the types of psychotropic medication provided to service users in the five districts. Access to regular supplies of psychotropic medications is a major challenge across all districts. Access to antiepileptics is, however, not a challenge as it is readily available in all districts. Generally, psychotropic medications are supplied by the GHS from the national medical stores. These are received at the regional medical stores for onward distribution to the district via the district hospitals. Each district's mental health focal person allocates the psychotropics to all health facilities that have a mental health professional. The allocation is usually based on the patient load; facilities with higher patient load receive a larger allocation.

**Table 5. Estimated treatment coverage for top-three mental health conditions by district in the most recent year (2020) by district.**

| MNS disorder | Tolon | | Bongo | | Ahanta West | | Asunafo North | | Anloga | | GBD study 2019* |
|---|---|---|---|---|---|---|---|---|---|---|---|
| | Popn prevalence (%) | Treatment coverage (%) | Popn prevalence (%) | Treatment coverage (%) | Popn prevalence (%) | Treatment coverage (%) | Popn prevalence (%) | Treatment coverage (%) | Popn prevalence (%) | Treatment coverage (%) | Popn prevalence (%) |
| Schizophrenia | 0.13 | 0.69 | 0.2 | 1.03 | 0.08 | 0.38 | 0.03 | 0.16 | 0.07 | 0.34 | 0.2 |
| Depression | <0.01 | <0.01 | 0.11 | 0.03 | 0.03 | <0.01 | 0.01 | <0.01 | <0.01 | <0.01 | 3.9 |
| Epilepsy | 0.45 | 1.04 | 0.25 | 0.58 | 0.09 | 0.23 | 0.05 | 0.12 | 0.05 | 0.12 | 0.6 |
| Overall coverage | | 0.58 | | 0.55 | | 0.21 | | 0.09 | | 0.15 | |

* Global Burden of Disease [19].

As reported by one of the respondents, medication is a challenge as one cannot be certain of its availability at any given time, even in the tertiary psychiatric hospitals. Also, because of delays in the medication supply chain, most of the medications that eventually reach the district hospitals are usually near their expiry date. The analysis further found many instances where mental health service users had to buy their medications from outside sources. In terms of financing, the assessment found that all the antidepressants, mood-stabilisers and antiepileptics are fully covered by Ghana's National Health Insurance Scheme (NHIS); however, they must be prescribed by a medical officer.

Based on our estimation (using the proportion of the district population with active health insurance), only one in two people in the general population in Tolon, Bongo, and Anloga would have access to mental health medication because of their active NHIS status if they were to become mentally ill. Comparatively, the proportions are much lower in Ahanta West and Asunafo North. These estimates compare closely (except for Ahanta West) with estimates from a recent study that reported active/current membership of 50% of the population in the Kassena-Nankana East and West districts of Ghana [21]. The implication is that at least half the population in each of these five districts who develop mental health conditions requiring medications may not have access to medication.

These challenges notwithstanding, the districts have developed various strategies to help improve their supply chains. For example, in some of the districts with committed leadership, medications are procured by the district hospital from the open market. In other instances, mental health officers pre-finance access to medications for clients from private pharmacies.

**Monitoring and evaluation.** Monitoring and evaluating the quality of mental health services provision is a major challenge in all five of the districts. The analysis showed that, apart from routine data from the DHIMS, support for supervision of mental health professionals is weak and unstructured. At the district level, the mental health focal person conducts supervision of colleague mental health professionals, but this is usually unstructured and constrained by budgetary challenges. The leadership of the Ghana Health Service mental health unit undertakes supportive supervisory visits, but these are not regular and the limited staff makes it difficult to supervise all districts. Routine data in the DHIMS can be accessed at the district level and are used to generate annual progress reports. The information on the DHIMS is mainly obtained using three reporting forms: (a) mental health client status: this collects data on a range of 33 variables on the status of inpatients and outpatients at the facility, disaggregated by sex; (b) mental health community report: this is designed to record information on the various mental health activities completed by the mental health teams at the various facilities that provide mental health services; and (c) mental health report: this entails all the major

**Table 6. Commonly used psychotropic medication availability by district.**

| Psychotropic medication group | Usual medications used | Tolon | Bongo | Ahanta West | Asunafo North | Anloga |
|---|---|---|---|---|---|---|
| Antipsychotics (po) | Risperidone | Yes | Yes | No | No | Yes |
| | Olanzapine | No | No | Yes | No | Yes |
| | Quetiapine | No | No | No | No | No |
| Antipsychotic depot | Fluphenazine decanoate | No | Yes | No | No | Yes |
| | Haloperidol decanoate | No | No | No | No | Yes |
| Antidepressants | Fluoxetine | No | No | No | No | Yes |
| | Amitriptyline | No | Yes | Yes | Yes | Yes |
| Anxiolytics | Diazepam | No | No | Yes | Yes | No |
| Mood stabilisers | Carbamazepine | No | Yes | Yes | Yes | Yes |
| Antiepileptics | Carbamazepine | No | Yes | Yes | Yes | Yes |
| | Phenobarbitone | Yes | Yes | Yes | Yes | Yes |

psychopathologies. The key indicators commonly reported from these three forms are summarised in Table 7. Many other indicators are also available on the DHIMS. For example, the community mental health report form reports on outreach programmes, home visits, visits to traditional and faith-based centres, number of patients found in shackles/chains, among other indicators. Challenges with data quality in terms of completeness and accuracy were noted in the analysis.

**Referral systems.** In terms of referrals, districts have in place a referral system that caters for both in referral (mental health patients coming from adjoining districts) and out referral (for services of specialists outside of PHC facilities in the district). For example, in 2020, the numbers of referrals in each district were: Asunafo North, in referrals: 7, out referrals: 13 (3.6% of total cases); Bongo, in referrals: 7, out referrals: 12 (1.5% of total cases); Tolon, in referrals: 9, out referrals: 5 (0.7% of total cases); Ahanta West, in referrals: 9, out referrals: 1 (0.8% of total cases); and Anloga, in referrals: 1, out referrals: 4 (1.8% of total cases). Within the districts, referrals can also be made by non-mental health professionals, who refer patients to mental health professionals in the same or another health facility in the district. There are systems for back referrals, but there is limited contact between district-level and specialist referral centres.

**Community support systems.** The analysis confirmed that traditional and faith-based healers (TFBHs) are the first port of call for many persons seeking mental healthcare across all districts (Table 8). However, the proportion of persons seen by TFBHs appears to vary across the districts; from less than 10% of patients in three districts (Tolon, Bongo, and Asunafo North), to a relatively higher patient base in Anloga (25% of patients) and Ahanta West (15% of patients). With these levels of patient base, the analysis showed that some TFBH centres

**Table 7. Commonly reported mental health indicators in the DHIMS.**

| Type of form | Reported indicators | Reporting frequency |
|---|---|---|
| Mental health client status | % of mentally ill with active NHIS membership by sex | Monthly |
| | % of attempted suicide cases by sex | |
| | % of suicide by sex | |
| Mental health community report | % of traditional and faith-based centres chaining mentally ill persons | Monthly |
| Mental health report | Per capita OPD attendance for mental health | Monthly |

**Table 8. Community support systems for mental health by district.**

| Indicator | Tolon | Bongo | Ahanta West | Asunafo North | Anloga |
|---|---|---|---|---|---|
| Community volunteers | 72 serving 172 communities with general health activities, including mental health | 246 serving 143 communities with general health activities, including mental health | 90 serving 123 communities with general health activities, including mental health | 170 serving 189 communities with general health activities, including mental health | 35 serving 99 communities with general health activities, including mental health |
| Mental health support groups/ self-help groups | Yes | Yes | Yes | No self-help groups | No self-help groups |
| | Groups meet regularly and invite mental health professionals to provide education | | | | |
| Help seeking for mental health | Most go to TFBH first | Health facilities are now the first port of call | TFBHs are usually the first port of call (about 90%) | Information unavailable | Clients visit TFBH centres before clinics |
| Traditional and faith-based healing sites | 11 TFBH centres | 15 TFBH centres | 67 TFBH centres | 3 TFBH centres | 4 TFBH centres |
| | Work collaboratively with mental health professionals to ensure client safety and provide psychotropics | Work closely with mental health professionals | Poor interaction with the MHPs; some attempts have been made to work with TFBHs but there is lack of interest | MHP working with only two. T; the others are not cooperative | Limited interaction with mental health professionals |
| Estimated % / number of mental health clients seen by TFBHs | About 8% all registrants in the most recent year | Less than 1% of all registrants in the most recent year | About 15% of all registrants in the most recent year access services of a TFBH | About 5% of all registrants in the most recent year access services of a TFBH | 25% of all registrants in the most recent year access services of a TFBH |
| | | | | The two TFBH sites currently have eight clients | |
| Family burden | Family burden is high; families are generally poor and unable to afford psychotropic medications | | | | |

attempt to collaborate with mental health professionals in the districts. However, Tolon and Bongo are the only districts where this collaboration has been successful in establishing a referral system that ensures patients have access to specialist care.

Two of the districts (Tolon and Bongo) also have active mental health/self-help support groups. The analysis also showed evidence that persons with mental health conditions receive support from government social interventions, but only in Bongo and Tolon. For example, the district directorate of health services has been able to collaborate with the Department of Social Welfare to get some people with mental health conditions and their families enrolled on the Government of Ghana's unconditional cash transfer programme, Livelihood Empowerment Against Poverty (LEAP) [22, 23], and registered on the NHIS (0.2% in Tolon; 5% in Bongo) for free. In terms of opportunities for awareness-raising and mental health literacy, all districts have well-established community volunteer systems, which support community mental health nurses in conducting outreach programmes. One of the districts (Tolon) has established mental health clubs within schools.

## Discussion

This paper describes a standardised situation analysis of mental health infrastructure and services in five districts in Ghana conducted to understand the PHC system and identify opportunities for integrating mental health and disability services. The findings confirm there is an unmet need for mental health services (<1% have access) amid poorly developed mental health systems and infrastructure. Across all districts, key information for estimating treatment coverage is unavailable, supervision of the few mental health professionals is weak and unstructured, access to regular supplies of psychotropic medications is a major challenge, and psychological treatments are extremely limited given the lack of trained clinical psychologists. Community support systems, however, appear encouraging, with evidence of a well-

established network of community volunteers and some collaboration with traditional and faith-based mental health service providers.

Ours is one of few studies of this nature conducted in an LMIC, but we observe that our findings compare closely with those of the PRIME situation analysis carried out across five LMICs (South Africa, Uganda, Ethiopia, Nepal, and India) [16]. For example, similar infrastructural deficits in terms of poor mental health staffing, weak supervision systems, and irregular psychotropic medication supply chains were observed in the five PRIME countries. The Ghana data, however, show a slightly encouraging community mental health support system, which did not feature strongly in any of the PRIME countries. The similarity of these findings serves to confirm reports that mental health infrastructure in LMICs is poor [24] and that there is a lack of quality mental health data for planning purposes [25]. The following sub-sections highlight both the challenges that are impeding mental health delivery and the opportunities and lessons for improving mental health service delivery in Ghana.

## Key challenges across districts

Across the five districts, we observe two key broad challenges in the provision of mental health services: implementation of the policy and legislative framework; and weak health systems for mental health and disability. We focus on these two challenge areas because of the central role they play in the provision of quality mental health services.

## Policy and legislative framework

Ghana is one of a small number of countries in Africa (less than 50% of countries in Africa [9]) with a mental health law. Ghana's Mental Health Act, 2012 (Act 846) is endorsed by the WHO as an example of best practice in mental health legislation, but the slowness of its implementation has affected governance and organisation of mental health services. Although the Act has facilitated the establishment of the MHA to govern mental healthcare in Ghana, essential drivers key to the promotion of primary mental healthcare are yet to come on line. For example, in the provisions of the Act (section 21), the MHA is supposed to establish District Mental Health Sub-Committees, which shall appoint District Mental Health Co-ordinators to be responsible for leading the implementation of mental health policy and activities; none of these were in place in the five districts in our study, or indeed anywhere in Ghana [26]. Poor governance is a barrier to effective integration of mental healthcare within PHC settings [7, 24]. As noted in the PRIME study, these governance structures are important for local-level planning of district services, to ensure they are responsive to local needs. There are challenges associated with the implementation of policy and legislation generally, and in the case of Ghana's Mental Health Act these were predicted at inception [27]. Stronger policy and legislation is needed, alongside Implementation research to help guide the process and ensure important bottlenecks are addressed before scale-up.

## Health systems for mental health

The challenges relating to policy and legislation have direct implications for mental health systems in the five districts studied. Our situation analysis findings for these districts were similar to those of the PRIME situation analysis: mental health structures and systems were poorly developed; there were no functional mental healthcare plans or dedicated budgets; human resource capacity, training, and supervision were inadequate; supplies of medications were unreliable; and there was a possible lack of competence in the use of psychological treatments. Other challenges include poor responsiveness of mental health information systems, as

evidenced by large gaps in available routine mental health service data. Addressing these challenges would be an essential part of a comprehensive mental healthcare plan.

Across all five districts, there is evidence of limited availability of mental health professionals, to the extent that there are no clinical psychologists or occupational therapists. This has resulted in the current practice where people living with severe mental illness (SMI) are managed by mental health nurses providing outreach visits at home, with support from the family. Ideally, a multi-disciplinary team should provide care for people living with SMI, especially at the acute stages. This unfortunately is not the practice on the ground as there is a significant shortage of specialist mental health professionals in these districts.

The numbers of mental health professionals are below those recommended by major regulatory bodies, such as the Canadian Psychiatric Association and the Royal Australian and New Zealand College of Psychiatrists. For example, Canadian Psychiatric Association recommends a ratio of one psychiatrist to 8000 population [28]. While such recommendations are useful for practice, it is worth noting that they are challenging to achieve, and they might not be universally applicable because of important normative considerations, such as context, needs, and local resources. Nonetheless, these human resource shortages were already known and had been reported in previous studies [16, 29]. Human resource challenges have consistently been found to impede the sustainable integration of adequate quality mental healthcare into PHC [30–32].

In terms of deficiencies in the mental health information system, Ghana is one of very few countries in Africa that has incorporated mental health data as part of the routine data collection at the district level, using the health information system branded as DHIMS [33]. A mental health information system is a system for collecting, processing, analysing, disseminating, and using information about a mental health service and the mental health needs of the population it serves. The current data processes suggest there is no real-time data entry on the DHIMS platform, and the modes of collection (via paper/improvised notebooks) and movement (from one facility to the other) pose significant challenges for data quality issues in terms of the key parameters of accuracy and completeness. For example, the 'non-specified mental health conditions' category would need to be reviewed to ensure accuracy in reporting. Beyond the issue of data accuracy, it was found to be difficult to access and report information on contact coverage for MNS conditions in the five districts. Key variables required to estimate this important service utilisation statistic (i.e. help-seeking behaviour and prevalence of mental health conditions) were not available. This finding is consistent with the findings of the PRIME situation analysis for other countries. Targeted interventions to improve how the mental health information system can be used to report service utilisation and mental health needs more accurately at the community level are needed. Future research in this area is encouraged.

## Opportunities for improved mental health services

The data from the situation analysis show there are several opportunities for improved mental health services across the five districts. As discussed, the challenges are undoubtedly considerable, but what is also discernible from the routine data and the interviews with key respondents is a resilient and robust district-level health system. We observe five key opportunities. First, having a committed leadership at the district level makes it possible to ensure regular supplies of psychotropic medications. Second, a supportive policy environment should ensure sustained committed leadership. We note the recent launch and on-going implementation of the 12-year mental health policy [34] and hope this will provide the needed impetus at the district level for the development of mental healthcare plans. Third, our study revealed some 'best practices' that have implications for integrated mental healthcare at the district level; for

example, all districts showed it is possible to include mental health services in maternal perinatal healthcare programmes, while some demonstrated this can also be applied to HIV programmes. These can be strengthened by leveraging existing guidelines from the MHA. Fourth, the elaborate DHIMS could be built on and positioned to collect accurate and complete data on a core set of mental health indicators for use in local and national planning. The use of standard mental health registers for collecting data (particularly at the community-level health facilities) and reporting at the unit level (as opposed to the current aggregated reporting) would be useful. This should enable unpacking of psychopathologies presently categorised as 'non-specified mental health conditions'. Finally, the role of community support systems, such as self-help groups and community-based volunteers, in mental healthcare featured strongly across all districts. These groups, together with the considerable collaborations with traditional and faith-based services, present a unique platform from which to explore modalities for community reintegration and rehabilitation services, and thus address the social determinants of mental health. This aligns with growing initiatives in global mental health that recognise the central role of community support systems in tackling these social determinants [35]. Ghana runs a relatively successful conditional and unconditional cash transfer programme, called Livelihoods Empowerment Against Poverty (LEAP) [22, 23, 36]. The Government also mandates all districts to dedicate a percentage (currently 3%) of funds allocated to the district to support persons with disability [37, 38]. These are both practical ways to support persons with mental health conditions, and strong community support systems can act as important catalysts in this regard.

## Strengths and limitations

Building on previous situation analyses, our study employed a relatively more rigorous methodology: we improved on the scope of a previous situation analysis in Ghana by adapting the comprehensive PRIME situation analysis tool to allow for the collection of additional important data on community support systems; we conducted a validation of the findings through a workshop with key respondents; and the data collection was done by an independent experienced consultant. Nevertheless, our study had some limitations. First, the analysis might not have captured an accurate picture of mental health services on the ground. For example, we might have underestimated the treatment coverage. This is because our data sources were problematic, particularly in terms of the accuracy and completeness of routine data. We dealt with this challenge through the report validation workshop, where participants provided critical clarification on the veracity of the data and sources. Second, our analysis was unable to include the perspectives of patients or service users directly because of logistical challenges. Third, we recognise that our findings might not be representative of all districts in Ghana, given that only five districts were involved in the study. We believe, however, that our rigorous selection process, which ensured we selected representative districts for the three geo-political zones of Ghana, enhances the generalisability of our findings.

## Conclusion

Across all five districts studied, the delivery of quality primary mental health services faces considerable challenges. The huge mental health service deficit (due in part to the lack of mental health professionals) could be tackled by providing PHC workers with the training and support needed to identify and treat mental health conditions. This has inspired Ghana Somubi Dwumadie to design, implement, and evaluate district mental healthcare plans in the next phase of this study, and the data from this situation analysis has been useful in selecting three

priority demonstration districts (Bongo, Asunafo North, and Anloga) that reflect a geographical balance.

## Acknowledgments

This study is an output of the Ghana Somubi Dwumadie (Ghana Participation Programme: info@ghanasomubi.com). We thank the following Ghana Somubi Dwumadie partners: Options (Sebastiana Etzo, Lyla Adwan-Kamara) for their roles in reviewing the situation analysis report presented at the validation meeting and for their comments on drafts of this paper. Thanks to Lutuf Abdul Rahman of Saha Consulting for his role in data collection. Thanks also to the district directors of health services for Bongo, Tolon, Asunafo North, Ahanta West, and Anloga.

## Author Contributions

**Conceptualization:** Benedict Weobong, Crick Lund.

**Data curation:** Kenneth Ayuurebobi Ae-Ngibise, Lionel Sakyi.

**Formal analysis:** Benedict Weobong.

**Funding acquisition:** Crick Lund.

**Methodology:** Kenneth Ayuurebobi Ae-Ngibise, Lionel Sakyi, Crick Lund.

**Writing – original draft:** Benedict Weobong.

**Writing – review & editing:** Kenneth Ayuurebobi Ae-Ngibise, Lionel Sakyi, Crick Lund.

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
