## [Decision Letter · Decision Letter 0]

1 Dec 2022

PONE-D-22-23817Towards implementation of context-specific integrated district mental health care plans: A situation analysis of mental health services in five districts in Ghana.PLOS ONE

Dear Dr. Weobong,

Thank you for submitting your manuscript to PLOS ONE. After careful consideration, we feel that it has merit but does not fully meet PLOS ONE’s publication criteria as it currently stands. Therefore, we invite you to submit a revised version of the manuscript that addresses the points raised during the review process.

ACADEMIC EDITOR:

The major issues with this manuscript are:

Methods are not described in sufficient details.

The manuscript needs to be presented in a more appropriate fashion. 

A thorough editing for English language will improve the overall presentation.

We look forward to receiving your revised manuscript.

Kind regards,

Bharat Bhushan Sharma, M.D.

Academic Editor

PLOS ONE

Journal Requirements:

2. Please note that in order to use the direct billing option the corresponding author must be affiliated with the chosen institute. Please either amend your manuscript to change the affiliation or corresponding author, or email us at plosone@plos.org with a request to remove this option.

Additional Editor Comments:

Reviewer1

The article is within scope of the journal and contributes to the body of existing evidence. Below are some few comments for authors to enhance the quality of the paper.

1. English editing will be of great value to this paper. There are long sentences. The background of the paper is not having a clear paragraph direction and therefore makes it difficult to determine the direction of the study. Some terms should be defined in the first use. E.g “the PRIME” study. The study's main aim is not well stated in the manuscript

2. The paper does not really follow a proper methodology reporting. This makes it difficult to gather information. It would be helpful to have sections where the study setting, study design, sampling procedures, data collection procedures and data collection instruments, inclusion and exclusion criteria, data analysis, etc. are clearly described. The current methodological draft is a mishmash of all of these sections in various combination.

Result- The type of data collected and the data collection procedures should be described in the methods section.

4. The discussion should be tightened up significantly - There are some text or statements without references. A concrete storyline is not visible. The discussion should be more succinct, with more literature supporting the argument/perspective the authors would like the reader to appreciate. It should clearly describe the social and policy implications of the study findings.

Reviewer 2

This manuscript is not written in Standard English and is not presented in an understandable manner overall. So, it needs to be revised and rewritten in a concise and clear manner.

As a limitation of this situational analysis, focused group discussions were not held with communities and beneficiaries of mental health services. This was a missed opportunity to explore their feelings and experiences in-depth regarding mental health disorders and the treatment options available in health facilities and the community.

Abstract: Despite having all scientific material, it is poorly written and needs to be thoroughly rewritten.

Background: It lacks information on other countries' trends regarding the integration of mental health services into non-specialized health facilities. It is better if other countries' trends are incorporated into the background section of this study.

Lines 131–133: The primary objective of this situational analysis is not in line with the aim stated in the abstract.

Methods: Line 138, "Methods, can be correct as "Methods and materials."

Line 140, "Design," and Line 143, "Setting," can be corrected as "Study setting and design." And it needs a summary in one paragraph.

In lines 143–145, you stated that the study sites were selected by key mental health stakeholders. That means the study sites were chosen on purpose by stakeholders; do you believe the districts chosen on purpose represent other Ghana districts?

The study period for this situational analysis is not clearly described within the section on "Methods and Materials."

Line 173-participants can be corrected as "secondary data source" and "study participants."

It is not clear how secondary data were collected, cleaned, and transferred to analysis.

Lines 177–179: insert a separated subtopic as "Ethical considerations."

And the rights of study participants were not clearly and specifically stated as ethical considerations.

Line 181, "Measures," can be corrected as "Data collection tools and procedures."

It is better to include "Data quality management" as separate sub-headings.

Line 202, "Data analysis," can be corrected to "Data processing and analysis."

Results: - Generally, the result section is too long and broad to read and easily understand within a short period of time. So, it needs to be summarized in a clear and brief manner.

Lines 214–222, you stated the source of data for this situational analysis. It is better if sources of data are only stated in the Methods and Materials section rather than in the Results section.

Line 225, "Socio-demographic and economic context," can be corrected as "Socio-Demographic and Socioeconomic Characteristics."

Line 246, "health profile," instead rewrite it as "health structure" or "health tire system."

Did Ghana have a mental health treatment guideline?

Did all health institutions located in the study district have mental health treatment guidelines?

Line 271, "mental health services organization," should instead be rewritten as "mental health police and regulation."

Lines 273-282 tell the story of Ghana's mental health authorities. Mental health authorities and offices are not mental health organizations, and their main responsibilities might be recruiting mental health professionals, planning mental health services, coordinating, planning, and conducting supportive supervision, delivering training for health providers, and conducting monitoring and evaluation. Since the Mental Health Act is not part of the study results, it would be better if Ghana’s Mental Health Act 2012 (Act 846) were incorporated into the background of this section of the study.

Lines 284–297 described mental health services plans, authorities, responsibilities, and roles at the regional and district level. What about mental health care plans at each and every level of mental health facilities (i.e., at health posts, health centers, district hospitals, and tertiary hospitals)? Similarly, this situational analysis did not report results showing health facilities' quarterly and annual plans or performances regarding mental health services at health facilities.

It is better if you summarize the two subtopics (availability of mental health services and human resources for mental health service provision) as follows: mental health professionals and mental health services at health facilities

You stated that people with mild and moderate mental and neurological disorders got mental health services at community health centers and district hospitals. What about people with severe mental illnesses who met all admission criteria and require close monitoring?

Line 384, table 3, description, needs revision. Please revise and rewrite it in a clear and brief manner.

The topic of the first column of table 3 was not appropriately written; revise it as "mental health activities at mental health offices and health institutions."

In Table 3 column 1, topic 2, you state "personnel type." Rephrase it as "mental health professionals from various backgrounds."

According to this situational analysis, alcohol detoxification was possible without institutionalization in Tolon, Asunafo North, and Anloga districts. How is alcohol detoxification possible without admitting the clients to health facilities? What are the researcher’s justifications?

You have reported that severe mental health illness is treated at home with the support of families in cases from all study districts. But it is difficult to treat SMI at the home level. According to the mhGAP intervention guidelines, all individuals with SMI should be treated by a psychiatrist at a psychiatry clinic or unit.So, what are your justifications for this inappropriate treatment service for people with SMI?

What are the parameters of mental health service coverage? What are the incidence and prevalence of mental, neurological, and substance use conditions in your study districts? In order to measure or estimate mental health services and coverage at study sites, incidences and prevalence of mental health disorders must be stated.Your situational analysis did not show the actual number of people with MNS conditions who had gotten treatment at health facilities in study districts. Furthermore, these situational analyses fail to provide a clear and concise description of the treatment gaps in the study area.

Lines 364-366 require revision because they are unclear and the write-up is inadequate.

Line 374-375, table 4, 19 words, which is more than the standards, needs revision.

Lines 384-385, able-5, 22 words, which exceeds the plos one journal standards

Line 387, subtopic "Availability and sources of psychotropic medication."

Similarly, the availability of a mental health plan, budget allocation, coordination, integration, and use of screening tools are not appropriately stated for each district's health offices.

Revise the paragraph as "psychotropic medication for people with MNS conditions" and try to summarize and rewrite the availability and constant supply of essential psychotropic drugs by kinds (i.e., anti-psychotics, antihistamines, anti-depressants, and anxiolytics) at each and every level of health facilities.

Table 6, last row and column 6, "Carbamazepine," is available at Asunafo North for epilepsy but not for mood stabilizers. Please, would you like to justify the reason behind these discrepancies?

Line 419, under subtopic "monitoring and evaluations," Did the district health offices and health facilities have clear plans to conduct monitoring and evaluation programs at each level of health facility? Did they have a well-established functional monitoring and evaluation team?

This paragraph under the subtopic "monitoring and evaluation" is not clear and concise enough to be read and understood. So, please summarize and rewrite this paragraph in a clear and brief manner.

Discussion: - lacks depth, is too shallow, and does not address other studies' findings from around the world. It is better to include "study limitations and strengths" as separate sub-headings.

Reviewers' comments:

Reviewer's Responses to Questions

**Comments to the Author**

1. Is the manuscript technically sound, and do the data support the conclusions?

Reviewer #1: Yes

Reviewer #2: Partly

2. Has the statistical analysis been performed appropriately and rigorously? 

Reviewer #1: Yes

Reviewer #2: No

3. Have the authors made all data underlying the findings in their manuscript fully available?

Reviewer #1: Yes

Reviewer #2: Yes

4. Is the manuscript presented in an intelligible fashion and written in standard English?

Reviewer #1: Yes

Reviewer #2: No

5. Review Comments to the Author

Reviewer #1: The article is within scope of the journal and contributes to the body of existing evidence. Below are some few comments for authors to enhance the quality of the paper.

1. English editing will be of great value to this paper. There are long sentences. The background of the paper is not having a clear paragraph direction and therefore makes it difficult to determine the direction of the study. Some terms should be defined in the first use. E.g “the PRIME” study. The study's main aim is not well stated in the manuscript

2. The paper does not really follow a proper methodology reporting. This makes it difficult to gather information. It would be helpful to have sections where the study setting, study design, sampling procedures, data collection procedures and data collection instruments, inclusion and exclusion criteria, data analysis, etc. are clearly described. The current methodological draft is a mishmash of all of these sections in various combination.

Result- The type of data collected and the data collection procedures should be described in the methods section.

4. The discussion should be tightened up significantly - There are some text or statements without references. A concrete storyline is not visible. The discussion should be more succinct, with more literature supporting the argument/perspective the authors would like the reader to appreciate. It should clearly describe the social and policy implications of the study findings.

Reviewer #2: This manuscript is not written in Standard English and is not presented in an understandable manner overall. So, it needs to be revised and rewritten in a concise and clear manner.

As a limitation of this situational analysis, focused group discussions were not held with communities and beneficiaries of mental health services. This was a missed opportunity to explore their feelings and experiences in-depth regarding mental health disorders and the treatment options available in health facilities and the community.

6. PLOS authors have the option to publish the peer review history of their article (what does this mean?). If published, this will include your full peer review and any attached files.

Reviewer #1: No

Reviewer #2: No

---

## [Author Response · Author response to Decision Letter 0]

25 Jan 2023

We thank you very for the thorough and patient review of our manuscript. We have prepared a point-by-point response for your consideration.

---

## [Editor Report · Decision Letter 1]

1 Feb 2023

PONE-D-22-23817R1Towards implementation of context-specific integrated district mental health care plans: A situation analysis of mental health services in five districts in Ghana.PLOS ONE

Dear Dr. Weobong,

Thank you for submitting your manuscript to PLOS ONE. After careful consideration, we feel that it has merit but does not fully meet PLOS ONE’s publication criteria as it currently stands. Therefore, we invite you to submit a revised version of the manuscript that addresses the points raised during the review process.

The manuscript has improved considerably after revision. However, it still needs a through copy-editing. Authors are requested to read the submission guidelines of the journal carefully and take help of a scientific editing service.

We look forward to receiving your revised manuscript.

Kind regards,

Bharat Bhushan Sharma, M.D.

Academic Editor

PLOS ONE

Additional Editor Comments (if provided):

The manuscript has improved considerably after revision. However, it still needs a through copy-editing. Authors are requested to read the submission guidelines carefully and take help of a scientific editing service.
---

## [Author Response · Author response to Decision Letter 1]

20 Feb 2023

Our rebuttal letter is uploaded as requested

---

## [Decision Letter · Decision Letter 2]

15 Mar 2023

PONE-D-22-23817R2Towards implementation of context-specific integrated district mental health care plans: A situation analysis of mental health services in five districts in Ghana.PLOS ONE

Dear Dr. Weobong,

Thank you for submitting your manuscript to PLOS ONE. After careful consideration, we feel that it has merit but does not fully meet PLOS ONE’s publication criteria as it currently stands. Therefore, we invite you to submit a revised version of the manuscript that addresses the points raised during the review process.

ACADEMIC EDITOR:

Please clarify following points raised by the reviewer and revise your submission accordingly:

Provide the financial grant number for your project/study.

Clarification needed regarding the treatment of severe mental illness at home by the patient's family.

We look forward to receiving your revised manuscript.

Kind regards,

Bharat Bhushan Sharma, M.D.

Academic Editor

PLOS ONE

Journal Requirements:

Reviewers' comments:

Reviewer's Responses to Questions

**Comments to the Author**

1. If the authors have adequately addressed your comments raised in a previous round of review and you feel that this manuscript is now acceptable for publication, you may indicate that here to bypass the “Comments to the Author” section, enter your conflict of interest statement in the “Confidential to Editor” section, and submit your "Accept" recommendation.

Reviewer #1: All comments have been addressed

Reviewer #2: All comments have been addressed

2. Is the manuscript technically sound, and do the data support the conclusions?

Reviewer #1: Yes

Reviewer #2: Yes

3. Has the statistical analysis been performed appropriately and rigorously? 

Reviewer #1: N/A

Reviewer #2: Yes

4. Have the authors made all data underlying the findings in their manuscript fully available?

Reviewer #1: Yes

Reviewer #2: Yes

5. Is the manuscript presented in an intelligible fashion and written in standard English?

Reviewer #1: Yes

Reviewer #2: Yes

6. Review Comments to the Author

Reviewer #1: (No Response)

Reviewer #2: Dear authors! You are doing a great job. Thank you for conducting this study. The majority of my concerns regarding this manuscript were well addressed, and you made the manuscript acceptable for publication. However, I still have a few concerns and comments on the financial grant number, the treatment of severe mental illness at home by the patient's family, and the recommendation.

7. PLOS authors have the option to publish the peer review history of their article (what does this mean?). If published, this will include your full peer review and any attached files.

Reviewer #1: No

Reviewer #2: No

---

## [Author Response · Author response to Decision Letter 2]

30 Mar 2023

Thank you for your further review of our manuscript. We have now prepared a point-by-point response attached to this resubmission.

---

## [Editor Report · Decision Letter 3]

20 Apr 2023

Towards implementation of context-specific integrated district mental health care plans: A situation analysis of mental health services in five districts in Ghana.

PONE-D-22-23817R3

Dear Dr. Weobong,

We’re pleased to inform you that your manuscript has been judged scientifically suitable for publication and will be formally accepted for publication once it meets all outstanding technical requirements.

Kind regards,

Bharat Bhushan Sharma, M.D.

Academic Editor

PLOS ONE

---

## [Editor Report · Acceptance letter]

9 May 2023

PONE-D-22-23817R3 

Towards implementation of context-specific integrated district mental healthcare plans: A situation analysis of mental health services in five districts in Ghana 

Dear Dr. Weobong:

I'm pleased to inform you that your manuscript has been deemed suitable for publication in PLOS ONE. Congratulations! Your manuscript is now with our production department. 

Kind regards, 

on behalf of

Professor Bharat Bhushan Sharma 

Academic Editor

PLOS ONE